# Hierarchical folding and reorganization of chromosomes are linked to transcriptional changes in cellular differentiation

James Fraser[1,†], Carmelo Ferrai[2,3,†], Andrea M Chiariello[4,†], Markus Schueler[2,†], Tiago Rito[2,†], Giovanni Laudanno[4,†], Mariano Barbieri[2], Benjamin L Moore[5], Dorothee CA Kraemer[2], Stuart Aitken[5], Sheila Q Xie[3,‡], Kelly J Morris[2,3], Masayoshi Itoh[6,7], Hideya Kawaji[6,7], Ines Jaeger[8,#], Yoshihide Hayashizaki[6], Piero Carninci[7], Alistair RR Forrest[7,¶], The FANTOM Consortium, Colin A Semple[5,*], Josée Dostie[1,**], Ana Pombo[2,3,***] & Mario Nicodemi[4,****]

## Abstract

**Mammalian chromosomes fold into arrays of megabase-sized topologically associating domains (TADs), which are arranged into compartments spanning multiple megabases of genomic DNA. TADs have internal substructures that are often cell type specific, but their higher-order organization remains elusive. Here, we investigate TAD higher-order interactions with Hi-C through neuronal differentiation and show that they form a hierarchy of domains-within-domains (metaTADs) extending across genomic scales up to the range of entire chromosomes. We find that TAD interactions are well captured by tree-like, hierarchical structures irrespective of cell type. metaTAD tree structures correlate with genetic, epigenomic and expression features, and structural tree rearrangements during differentiation are linked to transcriptional state changes. Using polymer modelling, we demonstrate that hierarchical folding promotes efficient chromatin packaging without the loss of contact specificity, highlighting a role far beyond the simple need for packing efficiency.**

**Keywords** chromatin contacts; chromosome architecture; epigenetics; gene expression; polymer modelling

**Subject Categories** Chromatin, Epigenetics, Genomics & Functional Genomics; Development & Differentiation; Genome-Scale & Integrative Biology
**Mol Syst Biol. (2015) 11: 852**

## Introduction

The spatial organization of chromatin in cell nuclei has essential functional roles. In mammals, chromosomes occupy distinct territories and have preferred radial positions that depend on cell type and transcription activity (Lanctot *et al*, 2007; Misteli, 2007; Bickmore & van Steensel, 2013; Tanay & Cavalli, 2013). Within chromosomes, chromatin is organized in megabase-sized regions, known as topologically associating domains (TADs), characterized by enriched levels of interactions (Dixon *et al*, 2012; Nora *et al*, 2012). TADs appear to contain inner substructures as revealed by high-resolution analyses (Sexton *et al*, 2012; Phillips-Cremins *et al*, 2013). At a larger scale, they generally fall into either compartment A or B, which are nuclear domains related to genomic function, up to tens of Mb in size enriched in active or repressed chromatin states, respectively (Lieberman-Aiden *et al*, 2009). Yet, the specificity of

1 Department of Biochemistry, Goodman Cancer Centre, McGill University, Montréal, QC, Canada
2 Epigenetic Regulation and Chromatin Architecture Group, Berlin Institute for Medical Systems Biology, Max-Delbrück Centre for Molecular Medicine, Berlin-Buch, Germany
3 Genome Function Group, MRC Clinical Sciences Centre, Imperial College London, Hammersmith Hospital Campus, London, UK
4 Dipartimento di Fisica, Università di Napoli Federico II, INFN Napoli, CNR-SPIN, Complesso Universitario di Monte Sant'Angelo, Naples, Italy
5 MRC Human Genetics Unit, MRC IGMM, University of Edinburgh, Edinburgh, UK
6 RIKEN Preventive Medicine and Diagnosis Innovation Program, Wako, Saitama, Japan
7 Division of Genomic Technologies, RIKEN Center for Life Science Technologies, Yokohama, Kanagawa, Japan
8 Stem Cell Neurogenesis Group, MRC Clinical Sciences Centre, Imperial College London, Hammersmith Hospital Campus, London, UK
‡Present address: Single Molecule Imaging Group, MRC Clinical Sciences Centre, Imperial College London, Hammersmith Hospital Campus, London, UK
#Present address: Cardiff School of Biosciences, Cardiff, UK
¶Present address: Systems Biology and Genomics, Harry Perkins Institute of Medical Research, Nedlands, WA, Australia
*Corresponding author. Tel: +44 131 651 8614; E-mail: colin.semple@igmm.ed.ac.uk
**Corresponding author. Tel: +1 514 398 4975; E-mail: josee.dostie@mcgill.ca
***Corresponding author. Tel: +49 30 94061752; E-mail: ana.pombo@mdc-berlin.de
****Corresponding author. Tel: +39 081 676475; E-mail: mario.nicodemi@na.infn.it
†These authors contributed equally to this work

TAD contacts within compartments A/B and how different structural levels of chromatin folding integrate with nuclear functions from the scale of individual genes up to the scale of chromosomes remain unclear. In particular, we lack a comprehensive understanding of the higher-order organization of TADs, the different scales to which TAD–TAD contacts extend, and how these higher-order structures change upon cell differentiation.

Here, we investigate higher-order TAD interactions in a neuronal differentiation model from mouse embryonic stem cells (ESC) via neural progenitor cells (NPC) to neurons. Novel analyses of Hi-C data sets during differentiation reveal that TADs form a hierarchy of domains-within-domains that we name "metaTADs". The metaTAD hierarchy extends across genomic scales up to the size range of entire chromosomes. We show that the complex inter-TAD interactions can be understood as relatively simple tree-like hierarchical structures irrespective of cell type. By comparing our Hi-C data with a variety of other data sets, we find that metaTAD tree structures correlate with patterns of epigenomic and expression features. Furthermore, the dynamics of tree rearrangements during differentiation link nuclear organization to transcriptional changes, providing a new paradigm to study chromatin structure and function. Using polymer modelling, we also demonstrate that hierarchical folding promotes efficient chromatin packaging without the loss of contact specificity. Our work highlights the close relationship between chromosome structure and function in mammalian nuclei, suggesting a functional role for hierarchical chromatin organization beyond simple chromatin packing efficiency.

## Results

To investigate higher-order chromatin folding during differentiation, we studied proliferating mouse embryonic stem cells (ESC), intermediate neuronal precursor cells (NPC) and post-mitotic neurons (Neurons; Fig 1A). ESC (46C cell line) were differentiated using a protocol optimized for large-scale production of functional murine neurons with a midbrain phenotype (Jaeger *et al*, 2011), and each time point showed homogeneous expression of stage-specific markers (Figs 1B and EV1A). Characteristic expression patterns for the cell types under study were also confirmed by genome-wide gene expression analyses by CAGE (cap analysis of gene expression) (Kodzius *et al*, 2006; Takahashi *et al*, 2012; Forrest *et al*, 2014) and Gene Ontology (Fig EV1B; Table EV1). Incorporation of bromodeoxyuridine (BrdU; 24 h) to mark cells undergoing DNA replication shows that while ESC and NPC are actively cycling, Neurons have ceased cell division (Fig 1C).

We produced Hi-C libraries for ESC, NPC and Neurons (Fig 1D), using a modified Hi-C protocol (Appendix Fig S1), which increases the yield of chromatin interaction products. Normalized Hi-C matrices show typical organization of chromatin into blocks of enriched interactions reflecting the existence of compartments and TADs (Fig 1D). This organization is chromosome specific, and we observe extensive changes during differentiation in the landscape of higher-order contacts of each chromosome (Appendix Figs S2 and S3). These patterns of structural dynamics often extend across the whole chromosomes and are accompanied by changes in genome-wide transcription activity in CAGE data that were produced from matched samples (examined in detail below). For instance, among the changes measured by CAGE, we find a quick depletion of the pluripotency transcription factors *Oct4* and *Rex1* after the ESC stage (Fig 1E). Similarly, we find that *nestin* and *Fgf5* are highly expressed in NPC, whereas the neuronal markers *Neurog2* and *Tubb3* are expressed in differentiated neurons (Fig 1E).

### TAD–TAD contacts extend across genomic scales to define higher-order structures

To investigate the architecture of higher-order chromosome folding, we first identified TAD positions across chromosomes in Hi-C data sets for all time points using the directionality index (Dixon *et al*, 2012) (Fig 2A, Appendix Fig S4, see Appendix Supplementary Analyses for details). For comparison, we also analysed a published Hi-C data set from a different mouse ESC line (ESC-J1; Dixon *et al*, 2012). Average TAD size was ~0.5 Mb across all cell types (Appendix Fig S5), consistent with recent reports (Phillips-Cremins *et al*, 2013; Pope *et al*, 2014; Rao *et al*, 2014). The location of TAD boundaries measured in our ESC-46C Hi-C data set and in the published ESC-J1 data set overlap by 83% (Appendix Fig S6), in the same range as the overlap typically reported between biological replicates (Dixon *et al*, 2012).

Although most chromatin contacts observed in Hi-C matrices are found within TADs, interaction signal is also detected locally between specific TADs (Fig 2A; Dixon *et al*, 2015) and extends to large genomic distances (Fig 1D). We explored higher-order contacts between TADs using Hi-C interaction matrices and found that the most frequently interacting partner of a given TAD is a flanking nearest neighbour TAD in 97% of cases. This behaviour points to a scenario where chromatin folds into larger domains containing multiple, preferentially interacting TADs.

To uncover the higher-order domain structure of chromosomes within Hi-C matrices, we implemented a single-linkage clustering procedure of Hi-C contacts. For each chromosome, we iteratively select the two most frequently interacting neighbouring TADs

**Figure 1. Chromatin contact maps (Hi-C) and matched gene expression (CAGE) data in murine neuronal differentiation system.**

A   Scheme of the murine differentiation system of our study, from ESC to NPC and post-mitotic neurons.

B   Cells express stage-specific markers as detected by immunofluorescence: ESC express Oct4, NPC the neuronal precursor marker nestin and Neurons Tubb3 (Tuj1 antibody). Scale bar, 100 μm.

C   ESC and NPC are actively cycling, whereas Neurons are negative for BrDNA after 24-h BrdU incorporation. Nuclei were counterstained with DAPI.

D   Examples of interaction patterns in Hi-C matrices across the whole chromosomes show extensive higher-order dynamic contacts, which change during terminal neuronal differentiation. Hi-C interaction data are plotted in log scale.

E   Matched CAGE data sets were produced from total RNA extracted from ESC, NPC and Neurons. The expression levels confirm specific expression of stage-specific markers. CAGE expression reported as a percentage relative to highest expression.

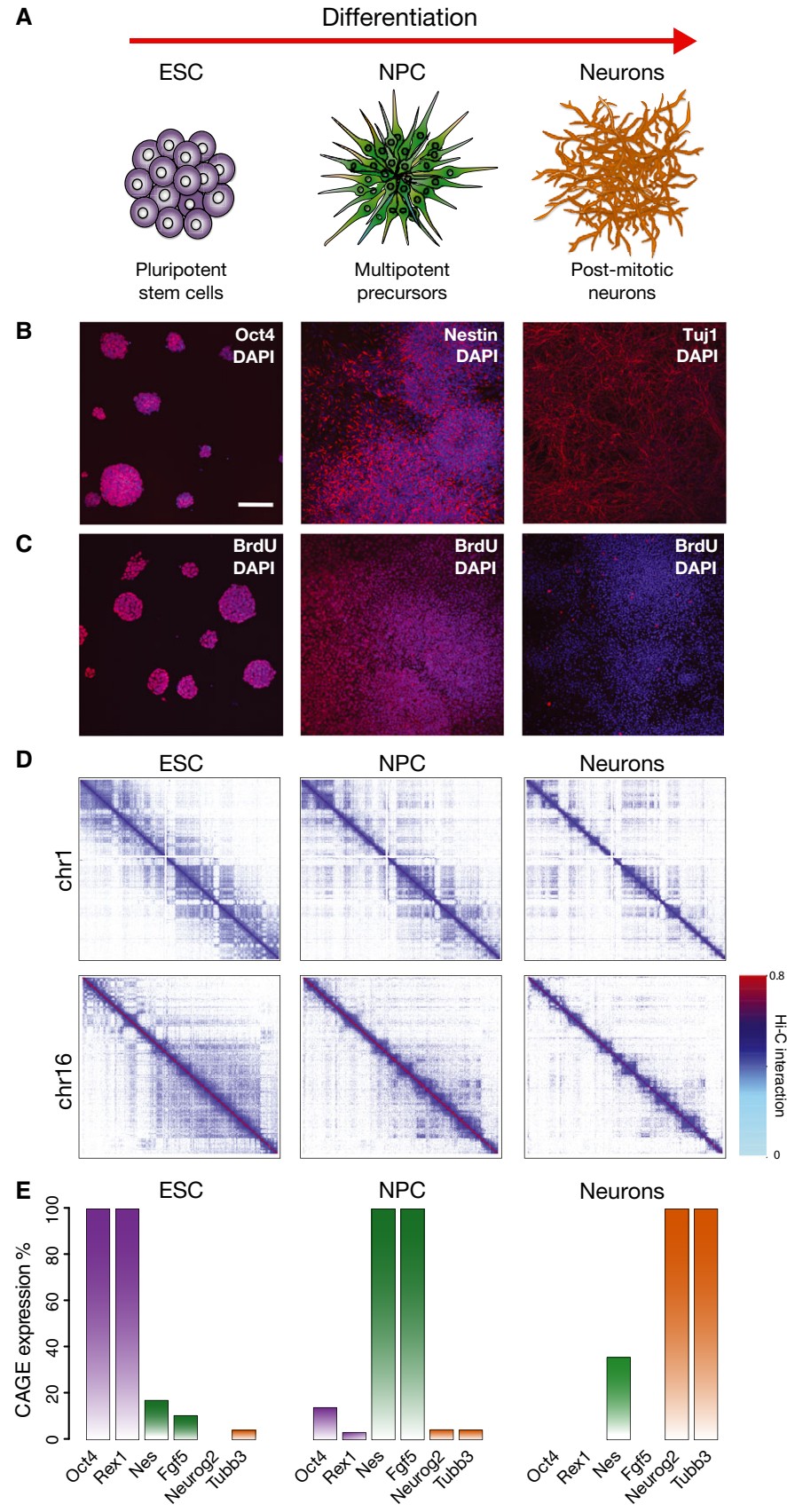

**Figure 1.**

(Fig 2B) and merge them into a higher-order domain or "metaTAD". This metaTAD is then added back to the list of domains, and the procedure is repeated until the entire chromosome arm is contained in a single domain that encompasses the hierarchy of all intervening lower-level metaTADs. The hierarchy of TAD–TAD contacts can be intuitively represented as a tree, where the joined domains are named sequentially as metaTAD-I, -II, -III and so on based on their position along the tree (detailed below). Overlaying the metaTAD structure onto Hi-C data (Fig 2C) provides a visual confirmation that the hierarchy of higher-order domains derived by single-linkage clustering matches the patterns of Hi-C data. It identifies the most frequent inter-TAD contacts in the first levels of the tree (I, II and III) and progressively lower Hi-C frequencies at the higher tree levels (IV and V).

To quantify the statistical reliability of the identified metaTADs, we examined several measures of inter-TAD contacts across all data sets. First, we tested whether interactions between metaTAD pairs are significantly more frequent than background interactions. We measured the average interaction level, $I$, between pairs of domains that produce new metaTADs containing a total of $n$ TADs. As background reference, we used the average interaction ($I_C$) between regions of the same genomic size, but randomly placed at the boundary of any other neighbouring TADs. In ESC, NPC and Neurons, the normalized interaction ratio $I/I_C$ remains significantly above control levels (measured in randomized Hi-C matrices), up to metaTADs containing several tens of TADs (Fig 2D). To give a sense of scale, we also plot $I$ at increasing tree levels to show the extent of Hi-C interactions between metaTADs (Fig EV2). $I/I_C$ remains 20% above the average values observed for randomized Hi-C, up to metaTADs containing ~80 TADs (Fig 2D, Appendix Fig S7A–D) corresponding to genomic lengths of ~40 Mb (Fig 2E). We also found that the normalized chromatin interactions detected *within* whole metaTADs, $J/J_C$, remain above background levels up to roughly the same length scale (Appendix Fig S7E–H; Fig EV2). As an additional control, we corrected the Hi-C data for 1D proximity effects (Appendix Supplementary Methods) and found the same most interacting TAD partners in 72% of cases, demonstrating that the observed metaTAD hierarchy is not only a consequence of linear distance. These analyses show that chromosomes adopt hierarchical structural conformations of increasing complexity of metaTADs in

ESC, NPC and Neurons, with prominent intra-TAD and inter-TAD contacts. These findings were fully confirmed using the original data sets of TADs identified in mouse ESC-J1 and in human IMR90 and ESC-H1 Hi-C data (Dixon *et al*, 2012) (Appendix Fig S8, Appendix Supplementary Methods).

Taken together, our results show that a hierarchical architecture of domains-within-domains is a general feature of chromatin folding, found across all stages of differentiation examined and in both murine and human cells.

### The metaTAD contact hierarchy bridges chromatin organization between TADs and nuclear compartments

To visualize the organization of chromatin in metaTADs at a genomic scale, we built a tree diagram for each chromosome, where the tree "leaf" nodes represent TADs and the internal nodes correspond to metaTADs (Fig 2B). Comparison of this type of diagram shows a visual correlation of the tree structures with A/B compartment domains, as tree sub-branches often coincide with transitions between compartments (Fig 2F). To test how metaTADs compare with compartments A/B, we measured the frequency with which two TADs in a common metaTAD are present in the same compartment (Appendix Fig S9). We found that TADs within a metaTAD frequently belong to the same compartment, in particular in the lower tree levels, as expected. Furthermore, this frequency is much higher than what is observed considering the linear distance between TADs, suggesting that the preferential contacts captured in the metaTAD hierarchy reflect preferential contacts within the same compartment type.

As an additional comparison between the metaTAD hierarchy of contacts and a second well-known (and independently measured) feature of chromatin organization, we studied the relationship between the TAD–TAD hierarchy and lamina-associating domains (LADs; Peric-Hupkes *et al*, 2010). Interestingly, we find that approximately half of the boundaries of metaTADs larger than 10 Mb coincide with LAD boundaries, a much higher fraction than expected by chance (Fig 2G, $P$-value $< 1 \times 10^{-4}$, see Appendix Supplementary Methods). Similar results were obtained using metaTADs larger than 5 or 20 Mb (Appendix Fig S10). These results reinforce the view that the metaTAD hierarchy of higher-order TAD contacts captures a

**Figure 2. Chromosomes are organized in a hierarchy of higher-order domains (metaTADs).**

A   ESC Hi-C map of chromosome 2, 53–58 Mb. The directionality index (DI, bottom) was used to identify TADs, numbered 1–6.

B   metaTAD identification by single-linkage clustering.

C   Examples of TADs (1–6) and metaTADs (I–V) in the same region shown in (A).

D   metaTADs are domains with enriched Hi-C contacts. The ratio of average interaction, $I$, between pairs of TADs or metaTADs, and background value, $I_C$, was calculated for ESC, NPC and Neurons, as a function of the total number of TADs ($n$) included in the metaTAD. $I/I_C$ remains 20% above control levels in randomized Hi-C matrices up to scales of the order of $n = 80$ TADs.

E   metaTADs size, $d$, is represented as a function of the number of TADs that they contain, $n$, showing that eighty TADs correspond to an average genomic length of around 40 Mb.

F   The metaTAD tree organization in ESC versus Neurons largely coincides with stretches of compartment A (grey) or B (black). A/B compartments are represented in the two central bars and were defined based on an individual principle component (green line) derived from Hi-C data. The yellow line indicates a value of 0.

G   Boundaries of metaTADs larger than 10 Mb are more enriched for transitions between lamina-associated (blue) and lamina-detached (red) regions than TAD boundaries. Heatmaps display the 900-kb flanking domain boundaries (dashed lines) for metaTADs (left heatmap) and TADs (right heatmap) for all boundary regions (heatmap rows). Transitions in lamina association are visible as abrupt changes in heatmap colours at boundaries (see Appendix Supplementary Methods). Both metaTAD and TAD boundaries are significantly more frequently observed to coincide with transitions than expected ($P < 1 \times 10^{-4}$; see Appendix Supplementary Methods).

H   The metaTAD tree of chromosome 19 in ESC (left: full; right: zoomed region). Interactions between metaTADs are not homogeneous, but instead occur through specific contacts involving specific TADs. Hi-C interaction data are plotted in log scale.

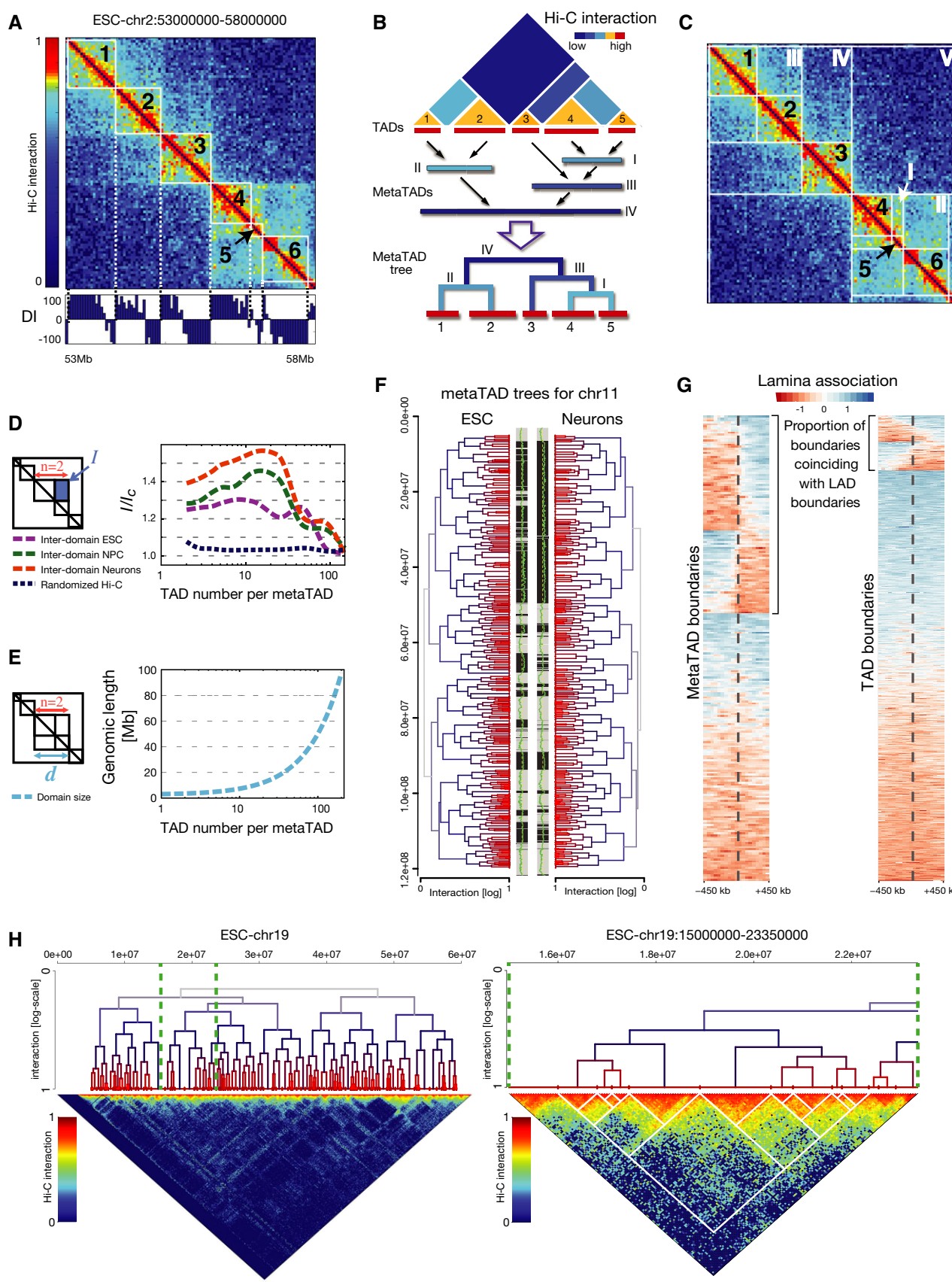

**Figure 2.**

complex series of architectural structures that bridge our understanding of higher-order chromatin folding between the (sub-mega-basepair) TAD scale and the A/B compartments or LADs, up to the scale of whole chromosomes.

Direct comparison of the trees of inter-TAD contacts with Hi-C contact matrices shows that the long-range interactions between metaTADs are not homogeneous, but instead involve only subsets of their respective inner TADs that often display preferential contacts with each other (Fig 2H). This suggests that the inter-metaTAD contacts likely occur between preferential TADs, and may reflect finer preferences in how chromatin folds that go beyond structural constraints imposed by compartments A/B or lamina association.

## The hierarchy of TAD–TAD contacts correlates with epigenomic features

To explore whether the hierarchical organization of metaTAD might reflect associations between chromatin domains with similar functional features, we investigated whether epigenomic features are more correlated across the metaTAD tree structures than their linear genomic neighbourhoods.

We started by studying how the hierarchy of metaTAD contacts identified in ESC relates with transcriptional activity and to a wide range of other genomic, epigenomic and expression features (Figs 3A and EV3A), using our matched CAGE data and publicly available ChIP-seq data sets mapped in mESC for histone marks, RNA polymerase II (RNAPII) phosphorylation state and sequence-specific transcription factors, including STAT3, Smad1, Zfx, c-Myc, n-Myc, Tcfcp2l1, E2f1 and the pluripotency factors Nanog, Oct4, Sox2, Klf4 and Esrrb (Mikkelsen *et al*, 2007; Chen *et al*, 2008; Hiratani *et al*, 2008; Peric-Hupkes *et al*, 2010; Encode Project Consortium, 2011; Brookes *et al*, 2012; Ferrari *et al*, 2014). We asked whether the functional features detected within TADs or metaTADs at a given tree distance are more correlated with each other (i.e. are more similar) than expected along the linear DNA sequence at the corresponding genomic distances (Fig 3B, Appendix Fig S11). Strikingly, we found that a number of features correlate over much longer genomic distances within topological contact trees than along the linear DNA sequence. These include replication timing, lamina association, histone marks associated with active transcription (H3K4me3, H3K27ac, H3K36me3), CTCF and several transcription factors expressed in ESC, RNAPII and CAGE transcription levels. Interestingly, we observe higher correlation lengths for RNAPII modifications that mark primed states of transcriptional activation (S5p), than for S7p that marks active promoters in the transition

to productive elongation, with the lowest correlation length for S2p, which marks elongation rates. The lower correlation length observed for S2p nicely corresponds to the lower correlation length observed with CAGE data, which independently measures active transcription. Considering that RNAPII-S5p marks poised states of genes expression, both at active and at Polycomb-repressed promoters (Brookes *et al*, 2012), it is also interesting to observe that RNAPII-S5p and H3K27me3 have similar, high correlation lengths, consistent with a role of Polycomb repression in chromatin architecture (Williamson *et al*, 2014). In contrast, other constitutive heterochromatin features, such as H3K9me3 and H4K20me3, are uncorrelated with the hierarchy of chromatin contacts (Appendix Fig S11), suggesting that local abundance of constitutive heterochromatin marks is unlikely to determine the higher-order topology of chromatin contacts. Additional genomic features used as control, such as the average exon count, were found to be uncorrelated with the tree structure, as expected.

As an additional control to test whether these correlations could happen simply due to clustering of genomic features along the linear genome sequence, we produced a collection of random trees where metaTADs are formed by randomly pairing neighbouring TADs (rather than the most frequently interacting neighbouring TADs or metaTADs). We found significantly lower correlations in the random trees than in the observed metaTAD trees (*P*-values obtained from comparing the two correlations are shown in Fig 3C, Appendix Fig S12), confirming that the TAD–TAD hierarchy of contacts captures long-range interactions between chromatin regions with similar epigenomic or expression features.

To test whether different epigenomic features may contribute to TAD–TAD contacts to different extents at different genomic distances, we investigated the genomic length scale across which different genomic or epigenomic features correlate. We calculated the correlation lengths for all features at a given correlation value (20%; set arbitrarily for comparison, as standard in this type of analysis) and found that different biological features have very different tree correlation lengths, which are typically much longer than their correlations over the linear (TAD) distance (Fig 3D). Interestingly, while histone marks associated with active transcription displayed longer tree correlation lengths, those linked to constitutive heterochromatin did not (Fig 3C and D, Appendix Figs S11 and S12). In agreement, we also found that transcription factors expressed in ESC, which include several pluripotency factors, are also enriched over longer distances in the trees than along the linear genome (Fig 3D, Appendix Figs S11 and S12). To test whether the correlations of the metaTAD trees with active chromatin were preserved upon neuronal differentiation, we measured the

---

**Figure 3.  metaTAD tree organization correlates with genomic, epigenomic and expression features.**

A   Comparison between the distribution of epigenomic, genomic and gene expression features with the metaTAD tree structure.

B   Left side: the diagram represents the difference between linear and metaTAD tree distance (number of edges along the tree minus one) for a given TAD (yellow) relative to other TADs (blue) in the same tree. Right side: correlations over the tree extend up to genomic scales of tens of Mb (filled circles) and are significantly stronger than those observed in linear genomic sequence (filled triangles). The horizontal dashed line indicates a 20% correlation coefficient.

C   Statistically significant differences are observed in the correlations measured across the metaTAD tree and across random neighbour trees constructed from the same linear array of TADs (horizontal line *P*-value = 0.05). Heterochromatin marks H4K20me3 and H3K9me3 levels do not correlate with the tree structure above what is expected from linear genomic distance.

D   CAGE data, different epigenomic features and pluripotency transcription factors binding sites (TFBS) have different average correlation lengths.

E   Genome-wide profiles averaged over all TAD or metaTAD boundaries of selected epigenomic features and gene densities (ribbons show 95% confidence intervals of the mean; metaTADs considered had length 10–40 Mb). Higher feature enrichments are observed at metaTAD boundaries (green) than TAD boundaries (violet).

F   Approximately two-thirds of TAD boundaries are conserved from ESC to Neurons.

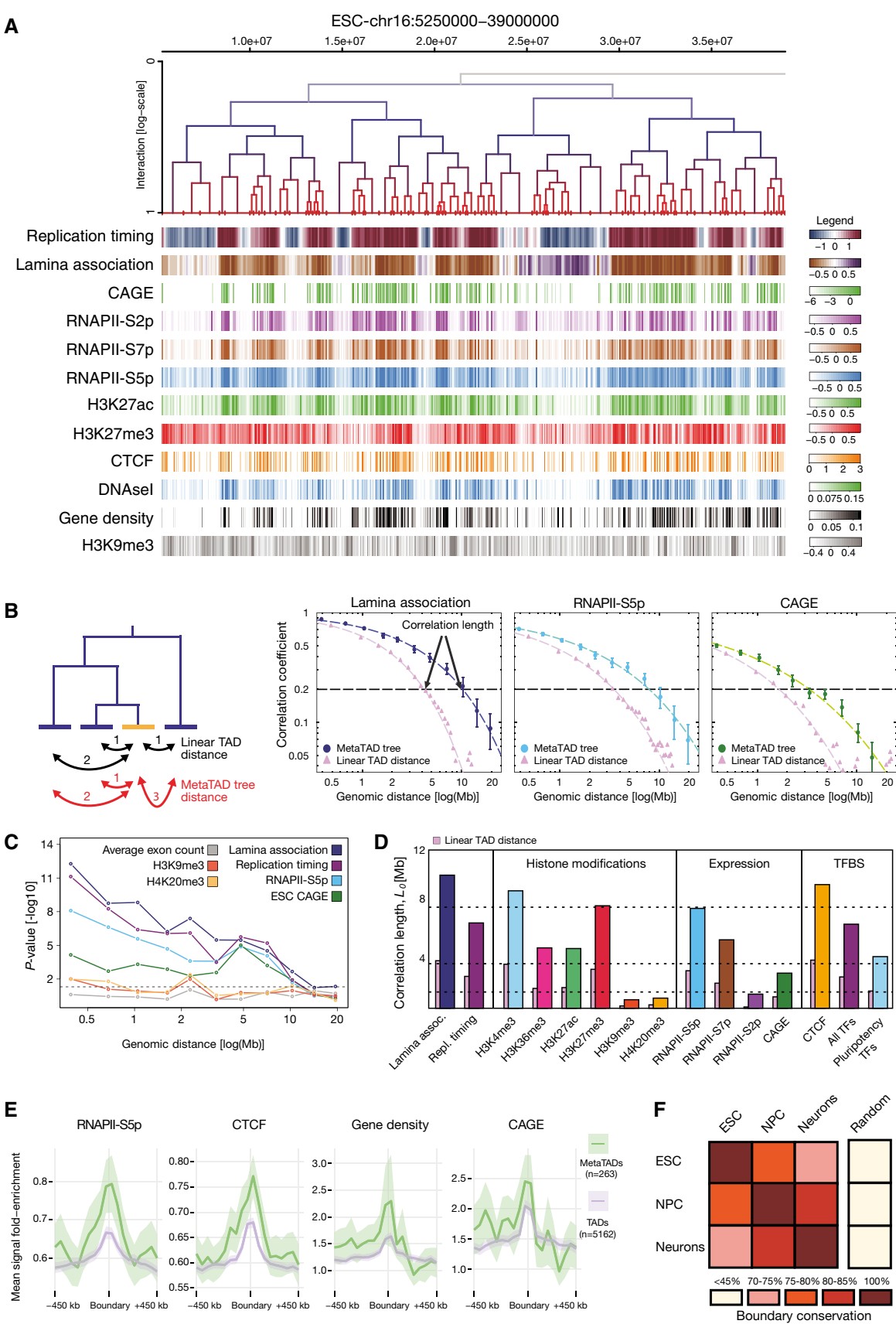

**Figure 3.**

correlation lengths for matched CAGE data and found strong correlations of CAGE transcript levels within the metaTAD trees of NPC and Neurons (Fig EV3B), despite metaTAD restructuring observed during differentiation (Fig 2F). These results suggest that different epigenomic features relate to, or may drive, TAD–TAD contacts at different genomic length scales in a combinatorial way. They also suggest that the hierarchical spatial organization of chromosomes into metaTADs is deeply connected to genomic functions, far beyond the simple need to tightly pack chromosomes in the nucleus.

The boundaries of TADs are enriched for specific genomic features (Dixon *et al*, 2012; Nora *et al*, 2012; Phillips-Cremins *et al*, 2013; Moore *et al*, 2015). The hierarchy of metaTADs identifies different types of domain boundary, comprising boundaries that connect two TADs at the lowest metaTAD levels up to boundaries that separate higher-order metaTADs containing large blocks of TADs. We measured the enrichment of chromatin features across TAD and metaTAD boundaries genome-wide, such as RNAPII and CTCF occupancy, and promoter activity measured by CAGE in ESC, NPC and Neurons (Fig 3E, Appendix Figs S13 and S14). Interestingly, we found that features previously observed as significantly enriched at TAD boundaries are even more strongly enriched at higher-order metaTAD boundaries (corresponding to genomic lengths of 10–40 Mb), consistent with important functional roles of the metaTAD organization.

### Hierarchical TAD–TAD contacts re-wire during terminal neuronal differentiation

To investigate the topological changes in the metaTAD trees during differentiation, we first examined the extent of changes in TAD boundary position across cell types. Conservation of TAD boundary position between ESC:NPC, NPC:Neurons and ESC:Neurons is 78, 80 and 74%, respectively, which is lower than observed with ESC biological replicates (Fig 3F, Appendix Fig S6). The decreasing trend as cells differentiate suggests a partial reorganization of TADs during differentiation.

We also sought evidence for structural reorganization of the metaTAD trees during differentiation. Using TADs with conserved boundaries, we compared the metaTAD topologies between time points using both global and local measures of tree structural changes (Fig 4A shows an example for the ESC–NPC transition on chr6). We first computed the *cophenetic* correlation, a global measure of tree similarity (see Appendix Supplementary Methods). Over all chromosomes, we found that the *cophenetic* correlation of tree topologies across differentiation stages is around 81–84%, well above random levels (Fig 4B and Appendix Fig S15).

Interestingly, comparisons of metaTAD trees for individual chromosomes show different degrees of tree re-wiring that depend both on the chromosome and on the time point transition considered. For instance, chromosomes 4, 6 and 19 have a global low tree similarity between ESC–NPC, but are relatively well conserved between NPC–Neurons (Fig EV4A). We conclude that the topologies of metaTAD trees undergo a degree of structural reorganization (~20%) during differentiation from pluripotent to terminally differentiated cells, against a background of substantial conservation.

In order to study how local reorganizations in metaTAD trees relate to changes in gene expression and to A/B compartment changes during terminal differentiation, we devised a measure to assess the extent of local tree changes surrounding each TAD between time points (Fig 4A heatmap). Briefly, we consider the neighbouring TADs of a given TAD (up to the distance of the third closest TAD over the tree) and compute their tree distance changes across two defined time points (Appendix Fig S16 and Appendix Supplementary Methods). We used this measure of local tree reorganization to classify TADs and identify genomic sequences undergoing significant local tree topology changes (employing a simple $z$-score $= 0$ threshold, Appendix Supplementary Methods). Overall, more TADs were classified as having local tree changes in the first, ESC–NPC transition, than between NPC–Neurons (Fig EV4B). Interestingly, the same trend was observed when comparing the overall length of genomic sequences corresponding to regions changing A/B compartment.

### Reorganization of metaTAD tree topologies overlaps with gene expression and compartment membership changes during differentiation

Next, we classified TADs according to their gene expression changes using the matched CAGE data produced for ESC, NPC and Neurons. To ensure a robust, per TAD, detection of gene expression changes, we considered both fold change and absolute CAGE signal differences, the latter to exclude TADs with large fold changes but very small absolute values of CAGE signal (e.g. genes with negligible, yet varying expression; Fig EV4C and Appendix Supplementary Methods).

We compared the positions of genomic regions exhibiting changes (or conservation) of the local tree structure with regions exhibiting changes in gene expression and/or changes in compartment A/B membership. We found that while some chromosome areas with tree changes display changes in gene expression, other genomic regions with conserved tree structures can also undergo gene expression changes (Fig 4C and Appendix Fig S16). Approximately 60% of TADs undergoing expression changes are associated with a change in compartment and/or tree structure (Fig 4D).

To assess whether the co-occurrence of gene expression changes with tree changes was non-random and suggestive of a functional relationship, we quantified genome-wide the overlap between genomic regions exhibiting gene expression changes and regions undergoing tree structure or A/B compartment changes. We used the Jaccard index, a measure of overlap between different intervals; for instance, a Jaccard index of 0.5 means that the length of overlap of two sets of intervals is half of their union. The statistical significance of the overlap values was assessed genome-wide by comparison with the overlaps seen with random circularly permuted regions of identical length (see Fig 4E for a scheme of the approach and Appendix Supplementary Methods for a more detailed description). We find a striking pattern of overlaps that are statistically significant and differ across the different cell transitions (Fig 4F) and which are robust to different criteria of gene expression changes (Appendix Fig S17).

A significant overlap between regions with both tree and gene expression changes is found in the ESC–Neurons transition, both in compartment A or in regions that undergo A/B compartment changes. This is also seen between ESC–NPC, but to a much lesser extent in the NPC–Neurons transition, suggesting that the early differentiation stages endure the most dynamic restructuring of TAD–TAD contacts in association with gene expression changes.

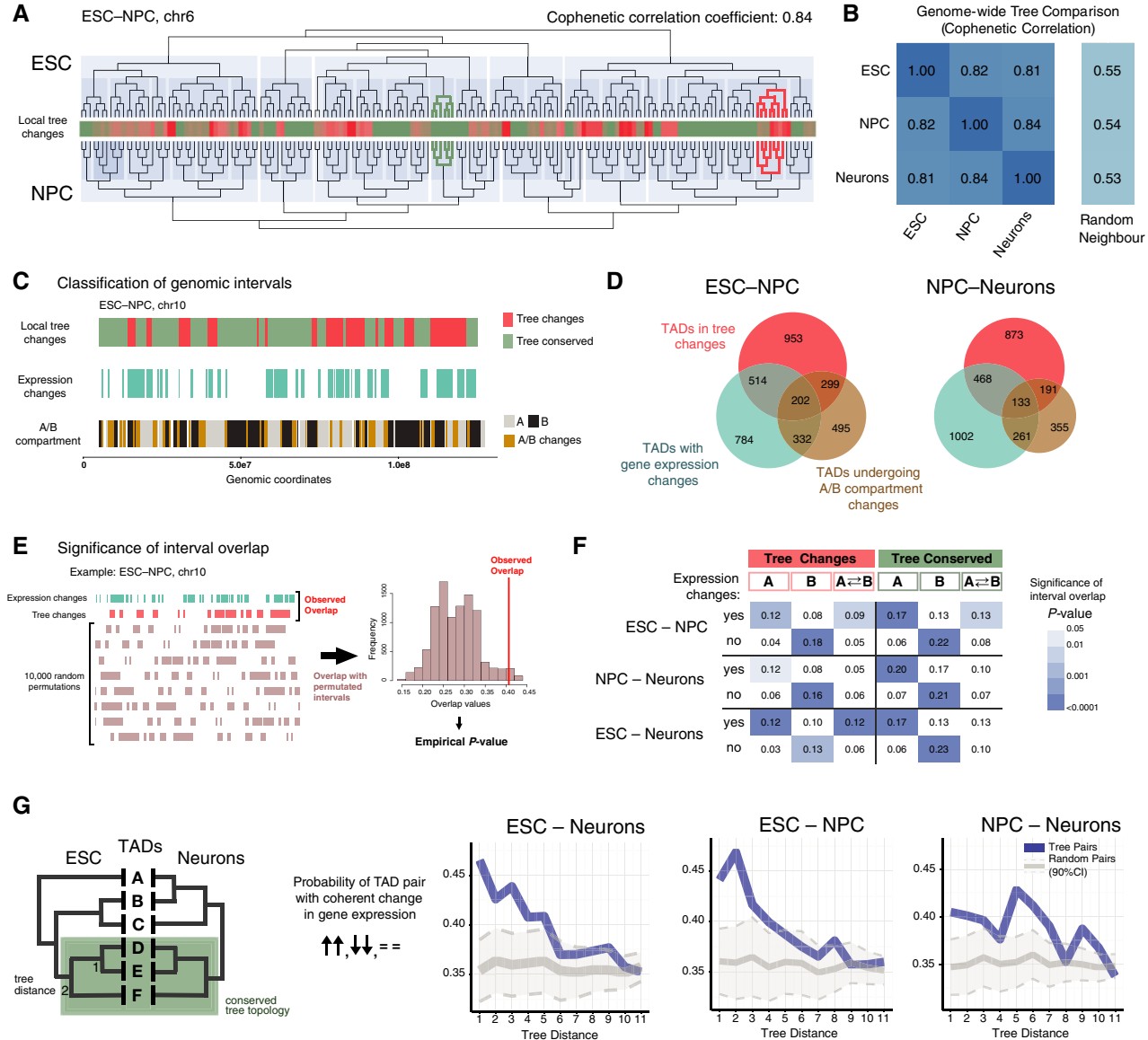

**Figure 4. Re-wiring of metaTAD trees during neuronal differentiation is linked to gene expression changes.**

A   Comparison between the metaTAD trees of chr6 in ESC and NPC. The extent of local tree topology changes is represented in the central heatmap. Examples of regions with conserved (dark green) and changing (red) local tree topologies are highlighted on the tree diagrams.

B   Cophenetic correlation coefficients between metaTAD trees at different time points show significant tree structure conservation (~80%) when compared to random neighbour trees.

C   Graphical representation of the distribution of tree changes, gene expression changes and A/B compartment membership changes across chr10 TADs for the ESC–NPC transition. Changes were computed at the level of conserved TADs.

D   Venn diagrams with the number of TADs that undergo changes in metaTAD tree topology, gene expression and A/B compartment in the ESC–NPC and NPC–Neuron differentiation transitions.

E   Scheme of the approach used to detect overlaps between genomic regions that undergo gene expression changes and tree changes or compartment A/B changes. The Jaccard index was used as a measure of overlap, and its significance was judged by comparing with 10,000 circularly permutated regions.

F   Overlap values (Jaccard index) between regions of gene expression changes and tree conservation/change, in compartment A, B or A-B transitions. The significance of overlaps is represented using a blue-coloured matrix; white cells represent non-significant overlaps (*P*-value > 0.05).

G   TAD pairs are more likely to display coherent changes in gene expression during differentiation (both TADs are upregulated, downregulated or unchanged) in regions where tree topology is conserved than random pairs. In metaTAD trees of TADs with conserved boundaries, this is significant up to a tree distance of 5 (corresponding to ~2–6 Mb in genomic distance).

Although some tree re-wiring can be detected within B compartment (Fig 4C), structural changes in contacts between TADs with persistent membership to compartment B are consistently not found

associated with gene expression changes. This suggests that tree re-wiring during differentiation within compartment B is more likely to happen to accommodate chromosomal structural changes (e.g.

due to different nuclear or nucleolar shapes) than gene expression changes. As a control, regions with no gene expression changes are invariably found in genomic intervals that always remain in B compartment (Fig 4F, "no" expression change rows). These results are consistent with the known functional properties of compartment B, which is generally transcriptionally repressed and enriched for heterochromatin (Lieberman-Aiden *et al*, 2009).

The observation of significant overlaps between tree changes and gene expression changes at all cell transitions (either in A compartment and/or involved in a compartment change) is consistent with a functional relationship between the metaTAD hierarchy and gene regulation where structural alterations might mediate expression changes, or vice versa. A relationship is also found between regions with structural tree conservation and expression change: comparisons between tree conservation, mostly in the context of compartment A, and gene expression changes identify statistically significant overlaps at all differentiation transitions (Fig 4F). Although it is less intuitive why gene expression changes could occur in the context of tree conservation regions, we reasoned that metaTADs might preserve their structure if the changes in gene expression within the encompassing TADs were coherent (i.e. both upregulated, downregulated or unchanged).

To test the hypothesis that metaTAD conservation can accommodate gene expression changes if they are coherent, we examined expression changes at TAD pairs retaining their metaTAD tree distance during differentiation (Fig 4G). Strikingly, we find that TAD pairs that maintain their tree distance between ESC and Neurons (conserved topologies) are enriched in coherent changes in gene expression, above random expectation, up to a tree distance of five in the metaTAD tree (Fig 4G, see Appendix Supplementary Methods for more details). Similar effects are observed between ESC–NPC, but the phenomenon is notably weaker in NPC–Neurons, perhaps because much of the neuronal gene expression programme is established before this latter transition. Nevertheless, it seems overall that chromatin domains that maintain their contacts during differentiation are more likely to display coordinated changes in gene expression, reinforcing the functional importance of higher-order metaTAD structures.

Taken together, our results show that changes in gene expression during differentiation are mostly found in compartment A, or in A-B/B-A compartment transitions. This pattern, interestingly, occurs both in areas of the tree that change local topology and, unexpectedly, in areas of the tree that maintain topology. The latter unexpected finding is explained by the observation that when the changes in gene expression within conserved metaTADs are coherent (in other words, when both TADs in a contacting pair coherently alter or maintain expression), they remain together in a conserved metaTAD (Fig 4G). Finally, regions of the genome that do not change expression tend to be and to remain in compartment B, although interestingly such regions can also re-wire their TAD contacts. A speculation is that such contact restructuring may be linked to topological re-wiring associated with accommodating gene expression changes occurring in adjacent domains.

## Polymer models of metaTADs predict mechanisms underlying metaTAD formation

To investigate the mechanisms underlying the formation of metaTADs and their influence on chromatin packaging, we used the strings and binders switch (SBS) polymer model (Barbieri *et al*, 2012; Nicodemi & Pombo, 2014), which explores chromatin folding mechanisms dependent on the formation of loops shaped by specific interactions with DNA-binding molecules (Fig 5A). Examining chromatin organization with first-principle polymer models has the potential to suggest physical mechanisms by which architectures may be formed. Although a few models have recently been proposed to explain how TADs are formed (reviewed in Nicodemi & Pombo, 2014), the SBS model was the first to recapitulate within a single framework a variety of data, from Hi-C and FISH and to explain TAD formation and chromatin looping (Barbieri *et al*, 2012; Nicodemi & Pombo, 2014). As a comparison, a simple self-avoiding walk polymer model or the fractal globule model would predict homogeneous Hi-C contact matrices with no TADs or metaTADs (Nicodemi & Pombo, 2014).

In the SBS model, a chromosomal region is represented as a polymer bearing binding sites for specific binders and the ensemble of polymer conformations is determined by computer simulations at thermodynamic equilibrium. We have previously shown that two different TADs can form spontaneously in a polymer containing two separate regions of different binding sites (Barbieri *et al*, 2012; green and red, Fig 5B). Here, we show that a metaTAD can be produced, for example, in a polymer where a third set of binding sites (blue) are interspersed with the existing red and green sites (Fig 5C). Strikingly, we find that the blue binding sites form a distinct spatial domain, inducing higher-order interactions between the two primary TADs, while the red and green sites are able to maintain their specific TAD structure. Other mechanisms can be envisioned to promote metaTAD folding, such as in the presence of distinct combinations of binding sites and factors. In our chosen scenario, the blue binding sites drive the formation of metaTAD structures and would be expected to contribute to correlations between epigenomic features observed at metaTAD scales (as in Fig 3B and D), and TAD pairs which both contain bridge-forming blue binding sites will be more likely to form a metaTAD structure than TADs with very different amounts of common (blue) binding sites.

To test the effect of the metaTAD formation on polymer packing, we compared the volume of the polymer and of its subcompartments (red, green and blue) in the two cases. Remarkably, by establishing relevant interactions between TADs, the blue binders reduced the average physical distance between the green and red TADs, resulting in a 50% reduction of the polymer volume (Fig 5D), while the red and green sites keep very high intra-domain contacts and vanishingly small binding across domains (Fig 5E). Interestingly, the blue binding sites only contact on 50% of the cases, as they are outcompeted by green–green or red–red binding, which are on average closer along the polymer. Thus, exploring the 3D properties of TAD–TAD interactions using the SBS model predicts that metaTAD formation has the potential to help to differentially segregate, insulate, contact or compact groups of genomic regions and thereby produce a variety of complex, yet dynamic and highly regulated chromosome conformations. Future work dissecting the roles of particular chromatin features associated with metaTAD structures will provide opportunities to test such predictions.

                    

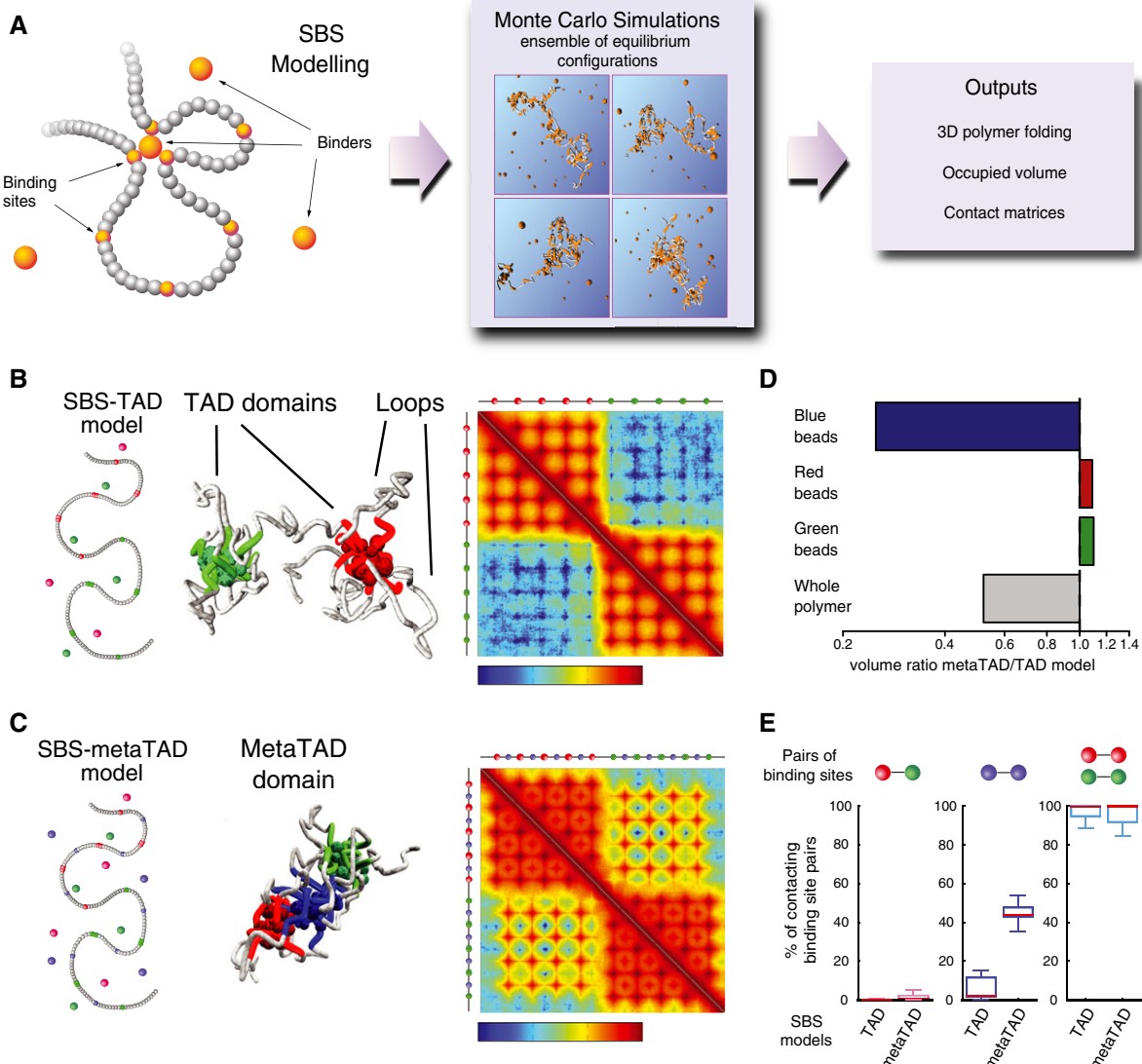

**Figure 5. Molecular mechanisms of metaTAD assembly.**

A  Schematic representation of the strings and binders switch (SBS) model. Binder particles form multiple bonds with binding sites on the polymer, allowing loop formation between distant regions. By Monte Carlo simulations, we produce equilibrium configurations to measure potentially relevant biological quantities, such as contact matrices, polymer compaction degree and other details of folding architecture.

B  The SBS model can explain mechanisms of hierarchical folding of TADs into metaTADs. Representation of an SBS-TAD model with two clusters of binding sites (red and green) and their respective binders, along with its emerging contact matrix (logarithm of contact frequency). The 3D conformation example shows the formation of loops and two separate domains.

C  Addition of a third type of interspersed binding sites (blue) and their binders promotes the formation of a metaTAD, as seen in the contact matrix (logarithm of contact frequency).

D  The volume of the polymer model represented in (C) (metaTAD model) is half the size of the model in (B) (TAD only model), albeit their length is the same. While the organization of red and green TADs remains unaltered, the formation of a third domain (blue) encompassing the blue binding sites results in increased proximity between the green and red TADs.

E  Despite increased compaction observed in the metaTAD model, the probability that a single red binding site contacts a green binding site in the same polymer remains vanishingly small. The contacts mediated through the blue beads result in a significant increase in the extent of polymer packing while keeping the red and green domains separated.

**Strong Hi-C signals detected between distant TADs correspond to close physical distances measured by cryoFISH**

Finally, to independently validate the existence of long-range contacts across several TADs and to investigate their distance and frequency of contact, we performed cryoFISH, a fluorescence *in situ* hybridization approach that combines the use of thin (~180 nm) cryosections for FISH and high-resolution imaging with confocal microscopy (Branco & Pombo, 2006). This method allows for optimal preservation of cellular architecture and the distribution of

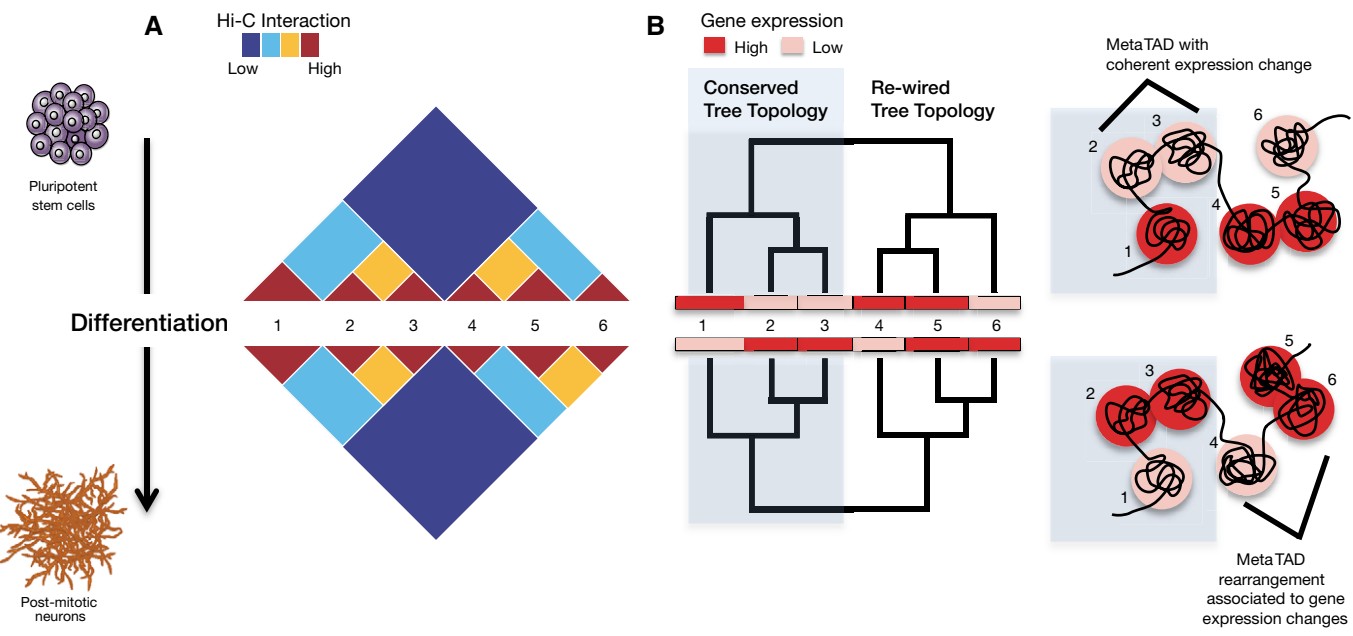

**Figure 6.   Hierarchical organization of chromosome folding in neuronal differentiation.**

A   Changes in chromatin contacts and gene expression were followed during differentiation of ESC into NPC and Neurons using Hi-C and CAGE. A hierarchy of higher-order contacts between TADs (numbered 1–6) was detected within Hi-C contact matrices, which is partially altered during differentiation. The patterns of TAD–TAD contacts reflect similarity in expression and epigenomic features.

B   The hierarchy of TAD–TAD contacts can be represented as a metaTAD tree which facilitates an integrated, multi-scale study of the relationship between chromatin contacts and expression and epigenomic features. Rearrangements in metaTAD tree topologies often occur at genomic regions that undergo gene expression changes, while conserved topologies often underlie TADs with coherent changes in gene expression.

subnuclear compartments. We designed three oligo libraries that cover three genomic regions in the mouse genome, regions *a* (red), *b* (green) and *c* (blue), belonging to distant, non-contiguous TADs on chr2, separated by 1.5 Mb (for *a-b*) and 2.0 Mb (for *b-c*; Fig EV5A). The probe positions were selected based on their different Hi-C interaction frequencies, to test whether the stronger interactions observed between regions *a* and *b* than regions *b* and *c* (~90% stronger) translate into closer spatial positioning at the single-cell level. FISH shows that regions *a* and *b* are separated by $350 \pm 242$ nm and are significantly closer than regions *b* and *c*, which are separated by $587 \pm 375$ nm (Wilcoxon test *P*-value 0.00168; Fig EV5B and C; the ratio of physical distances between *b-c* and *a-b* is 30% larger than the ratio between their genomic distances). These results support the view that strong Hi-C interactions identified between distant TADs form higher-order spatial structures.

## Discussion

Overall, the present work unveils a picture of chromatin architecture where each chromosome is folded in a hierarchy of TAD–TAD contacts (metaTADs) extending across spatial scales in a tree-like, hierarchical organization (Fig 6). metaTADs capture much of the residual interaction signal observed in Hi-C matrices after TAD detection. Just as in other systems, such as metabolic (Ravasz *et al*, 2002) and regulatory (Yu & Gerstein, 2006) networks, the hierarchical topologies of metaTAD trees are likely to play important functional roles and affect our views of genome evolution and

disruption in disease. We find that successive levels of metaTAD organization correlate with key genomic, epigenomic and expression features, providing an intuitive explanation for the genome-wide patterns of these features. Moreover, we find that metaTAD organization is evident in all considered cell types and in both mouse and human. These hierarchical topologies thus appear to reflect a general organizational principle of genomes, perhaps endowing critical features, such as local adaptability to specific stimuli amid broader structural stability. A hierarchical topology may facilitate chromatin compaction while retaining contact specificity or could be used to efficiently access and activate or silence a specific genomic region. Progress in our understanding of chromatin architecture, its mechanisms and functional implications, will also advance our ability to predict the broader consequences of local genomic rearrangements associated with diseases, such as genomic deletions, duplications or inversions in cancer and congenital disease (Crutchley *et al*, 2010; Spielmann & Mundlos, 2013).

## Materials and Methods

A more detailed description of the materials and methods is provided in the Appendix.

### ESC culture and neuronal differentiation

Mouse ESC (46C cell line) were grown as previously described (Abranches *et al*, 2009). Neuronal differentiation was done following Jaeger *et al* (2011), with modifications for large-scale culture.

Experimental details, including the incorporation of BrdU, immunofluorescence and microscopy, can be found in the Appendix.

## CAGE

CAGE libraries were prepared and sequenced on the HeliScope Single Molecule Sequencer platform (Helicos Bioscience). Sequencing reads were mapped and clustered into TSS regions, as previously described (Forrest *et al*, 2014).

## Generation of Hi-C libraries, sequencing and Hi-C data processing

Hi-C products and paired-end libraries were prepared as previously described (Lieberman-Aiden *et al*, 2009), with modifications that increase DNA product yield. Libraries were sequenced on the Illumina Hi-Seq 2000 platform and reads mapped to mm9, prior to normalization using iterative correction (Imakaev *et al*, 2012) and additional background subtraction.

## Bioinformatic analysis of metaTADs

Bioinformatic analysis for TAD identification, TAD boundary comparison, metaTAD identification, correlation with epigenomic and genomic features, correlations on random trees, analysis of emerging and conserved TADs and metaTADs, comparison of Hi-C trees and the relationship between gene expression and tree topologies are described in the Appendix.

## The strings and binders switch (SBS) polymer model

The SBS model was applied as previously described (Nicodemi & Prisco, 2009; Barbieri *et al*, 2012). Analysis details including SBS parameters, Monte Carlo simulations, contact matrices, gyration radius and occupied volume can be found in the Appendix.

## cryoFISH

Fluorescence *in situ* hybridization in thin cryosections was performed according to previous procedures (Branco & Pombo, 2006), except for the use of directly labelled MYtags oligo libraries (MYcroarray, Ann Arbor, MI, USA). Images were collected on a Leica SP8 confocal laser scanning microscope, before distances between FISH signals were measured using an automated macro. Detailed experimental procedures are described in the Appendix.

## Data availability

Raw reads for the ESC, NPC and Neuron Hi-C data sets generated for this study are available online from GEO, accession number GSE59027. CAGE data used in this study were produced as part of the FANTOM5 project, and all FANTOM5 sequence data have been deposited at the DNA Data Bank of Japan (DDBJ) under accession numbers DRA000991, DRA002711, DRA002747 and DRA002748. Additional analysis, documentation and visualizations of the CAGE data are available at http://fantom.gsc.riken.jp/5/tet/ under "ES-46C embryonic stem cells, neuronal differentiation, day00, biol_rep1.CNhs14104.14357-155I1" (ESC rep1);

"ES-46C embryonic stem cells, neuronal differentiation, day00, biol_rep2.CNhs14109.14362-155I6" (ESC rep2); "ES-46C derived epistem cells, neuronal differentiation, day05, biol_rep1.CNhs14126.14378-156B4" (NPC) and "ES-46C derived epistem cells, neuronal differentiation, day14, biol_rep1.CNhs14127.14379-156B5" (Neurons). The Tables EV2 and EV3 are discussed in the Appendix.

**Expanded View** for this article is available online.

## Acknowledgements

We thank Hisashi Miura for discussions on the preparation of Hi-C libraries, Meng Li for advice about neuronal differentiation, Elena Torlai Triglia and Giulia Caglio for initial processing of published ChIP-seq data sets and Robert A. Beagrie for critical discussions and comments on the manuscript. This work was supported by grants to JD from the Canadian Institutes of Health Research (CIHR) [MOP-86716, CAP-120350]. JD is a CIHR New Investigator and FRSQ Research Scholar (Fonds de la Recherche en Santé du Québec). AP and CF thank the BBSRC (UK); AP, CF, BLM, SA, SQX, KJM and CAS thank the Medical Research Council (UK); AP, CF, MS, TR, MB, DK and KJM thank the Helmholtz Foundation (Germany) for support. MN acknowledges computer resources from *Scope* and CINECA ISCRA HP10CYFPS5. FANTOM5 was made possible by a Research Grant for RIKEN Omics Science Center from MEXT to YH and a grant of the Innovative Cell Biology by Innovative Technology (Cell Innovation Program) from the MEXT, Japan, to YH. It was also supported by Research Grants for RIKEN Preventive Medicine and Diagnosis Innovation Program (RIKEN PMI) to YH and RIKEN Centre for Life Science Technologies, Division of Genomic Technologies (RIKEN CLST (DGT)) from the MEXT, Japan.

## Author contributions

JD, AP and MN designed the studies; CF, JF, DCAK and KJM conducted the experiments; SQX helped with FISH experiments; JF, AMC, MS, TR, GL, MB, BLM, DCAK, SA and CAS carried out the data analysis; IJ helped with the neuronal differentiation; ARRF, MI, HK, PC and YH generated the CAGE data as part of the FANTOM5 project phase 2; CF, MS, TR, JF, AMC, CAS, JD, AP and MN prepared the manuscript.

## Conflict of interest

The authors declare that they have no conflict of interest.

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
