## [Review Process File · Molecular Systems Biology]

Hierarchical folding and reorganization of chromosomes are linked to transcriptional changes in cellular differentiation

James Fraser, Carmelo Ferrai, Andrea Chiariello, Markus Schueler, Tiago Rito, Giovanni Laudanno, Mariano Barbieri, Benjamin L. Moore, Stuart Aitken, Sheila Q. Xie, Kelly J. Morris, Masayoshi Itoh, Hideya Kawaji, Inês Jaeger, Yoshihide Hayashizaki, Piero Carninci, Alistair R. R. Forrest, the FANTOM Consortium, Colin A. Semple, Josée Dostie, Ana Pombo and Mario Nicodemi

Corresponding authors: Colin A. Semple, MRC IGMM; Josée Dostie, McGill University; Ana Pombo, Berlin Institute for Medical Systems Biology and Mario Nicodemi, Università di Napoli Federico II

Review timeline:

Submission date:	08 August 2015
Editorial Decision:	21 August 2015
Revision received:	11 November 2015
Editorial Decision:	24 November 2015
Revision received:	26 November 2015
Accepted:	27 November 2015

Editor: Maria Polychronidou

Transaction Report:

1st Editorial Decision

21 August 2015

Thank you again for submitting your work to Molecular Systems Biology. We have now heard back from two of the three referees who agreed to evaluate your manuscript. We are still expecting the third report, but considering that both referees are cautiously supportive, I think that it is seems reasonable to make a decision now in order to save time. When reviewer #3 sends his/her review, we will forward it to you, so that you can address any additional issues raised. As you will see below, reviewers #1 and #2 acknowledge that you address a timely topic and think that the presented findings seem interesting. They raise however a series of concerns, which should be carefully addressed in a revision of the manuscript.

Without repeating all the points listed below, some of the more fundamental issues are the following:

- metaTADs need to be validated using an independent method (i.e. FISH).
- Some analyses of potential "binders" (as suggested by reviewer #2) could be included.
- Statistical support for the presented findings and information on the related statistical analyses need to be provided.

- Moreover, reviewer #1 refers to the need to include additional controls.

- Reviewer #1 recommends providing further experimental evidence demonstrating the functional relevance of the metaTADs. We have circulated the reports to both reviewers as part of our 'pre-decision cross-commenting' policy. During this process, reviewer #2 mentioned: "Reviewer #1 has several good points. In particular, I agree with: i) the suggestion to include 'active' histone marks and to discuss apparent contradictions in the epigenome comparisons, ii) suggestions to provide additional information on statistical methods, controls and significance iii) validation of metaTAD organization by FISH. However, I do not think it is necessary to do CRISPR experiments to (attempt to) disrupt metaTAD structure. It is unclear at this point what should be mutated. This problem is likely to keep the field busy in the next five years. It may take even longer before the functional relevance of (meta)TADs is solved."

As such, while further experimental analyses of the functional relevance of metaTADs would indeed significantly enhance the impact of the study and we would welcome their inclusion (i.e. if available), we do not think that they are mandatory for the acceptance of this work.

If you feel you can satisfactorily deal with these points and those listed by the referees, you may wish to submit a revised version of your manuscript. Please attach a covering letter giving details of the way in which you have handled each of the points raised by the referees. A revised manuscript will be once again subject to review and you probably understand that we can give you no guarantee at this stage that the eventual outcome will be favorable.

REFEREE REPORTS

Reviewer #1:

Fraser and colleagues in this manuscript introduce the notion of metaTADs, defined as higher order aggregations of TADs. They show that this organization occurs in a hierarchical fashion extending over several levels of folding and that metaTAD formation is driven by functional similarity between TADs and correlates with numerous features including RNA polymerase II presence, epigenetic modifications, CTCF presence and replication timing. Polymer modeling is finally used to demonstrate the feasibility of forming metaTADs by introduction of a class of binding sites in interacting TADs.

The notion of metaTADs is in all likelihood correct. It is very much what is expected based on what we know about chromatin fiber folding at this point. However, the data to support the existence of and to characterise metaTADs as presented in this manuscript are wanting. The manuscript a) lacks validation of the existence of metaTADs by independent methods, b) does not provide evidence, beyond correlation, of functional relevance of metaTADs, c) provides no indication of mechanism/drivers of metaTAD formation and d) controls and statistical analysis are missing.

The authors should provide experimental evidence for at least one of the major shortcomings (lack of validation by an independent method, lack of evidence for functional relevance, lack of mechanism).

Specific points:

The authors should provide a sense of scale for the frequency of TAD-TAD interactions at each level of the tree. The long range interactions are not well visible in the maps. Please provide frequencies of interactions at each level. How frequent are TAD-TAD interactions relative to intra-TAD interactions, how frequent are second level metaTAD-metaTAD interactions etc.?

The authors correlate metaTADs with numerous chromatin features. It is difficult to know which ones of these, if any, are relevant and are the drivers, as the authors suggest. Some seem to contradict each other (presence of active RNA pol II and H3K27methyl3). These correlations neither resolve the question of whether metaTADs are functional nor what their mechanism of formation is. The meaning of these various correlations remains unknown.

It is surprising that the authors only tested for a single histone modification (H3K27methyl3). Additional ones should be analyzed. Both correlating and non-correlating ones should be shown; particularly modifications related to transcriptional activity should be tested. Statistical analysis of their distribution relative to metaTADs should be provided.

The conclusion that metaTADs are functionally important relies entirely on correlation. Can the authors disrupt metaTADs, for example, by altering some of the chromatin features they appear to correlate with (I realize the proper experiment of CRISPR disruption will probably be deemed "beyond the scope of this study"). In the absence of direct functional testing by experimentation, functional claims are not valid and should not be made.

The authors should validate metaTADs by an independent method such as FISH and show differential pairing in ESC, NPC and neurons as proposed in figure 6, including of genes which change activity during differentiation and switch metaTADs as suggested by the authors in figure 6.

The authors find conservation of TADs between 74-80% in pairwise comparisons between ESC, PNC and neurons. How does this compare to an entirely unrelated cell line, say fibroblasts? Is this level of conservation functionally important?

"A significant overlap between regions with both tree change and gene expression change is found in the ESC-Neurons transition, both in compartment A or regions that undergo A/B compartment changes.". How is significance determined in this analysis? What are the controls? What are the statistical methods used and what are the values of statistical comparators?

The authors use the argument that genes which change expression during differentiation are found in changing metaTADs. Please provide comprehensive controls of genes which do not change expression during differentiation.

There is a general lack of information on the statistical analysis used. Numerous statements regarding significance and differences are made without controls or indication of statistical power. Please go through the manuscript and provide indicators of statistical significance of all claimed differences.

Minor:

The authors state that interaction strength is measured (p. 6). This is not the case. What is measured is an interaction signal. What this signal means is not known or tested. Please rephrase.

With regards to nomenclature, would it be useful to introduce the notion of metaTAD-I, metaTAD-II etc. to indicate where in the tree a metaTAD is.

The participating authors from the FANTOM consortium should be listed somewhere.

Reviewer #2:

This is an interesting and timely manuscript that addresses the hierarchical organization of Topologically Associated Domains (TADs). This hierarchical ("metaTAD") organization has been noted before, but to my knowledge has not been captured in a computational framework yet. The authors essentially use hierarchical clustering for this purpose. Perhaps the most interesting finding is that different levels of the hierarchy correlate with different chromatin features. In addition, the authors propose a model (supported by some simulations) - essentially an extension of their previous SBS model - to explain metaTADs. This last part remains highly speculative, but it is still worth considering.

Overall, this is an interesting manuscript that will fit well in MSB. I have the following comments and suggestions:

MAJOR:

SBS modeling (Figure 5): interesting, but it remains very theoretical. A key question is: what are the binders? Here I wonder whether an opportunity has been missed. Figure 3 only compares metaTAD architecture to chromatin marks that are unlikely to be binders. But what about DNA-binding factors and their sequence motifs (e.g. Jaspar database): do any of these show a TAD/metaTAD enrichment pattern? For a few human cell types there is a lot of ENCODE data as well as Hi-C data to test this.

In addition: Surely the SBS model is not the only model that may explain metaTADs. Needs to be addressed in the Discussion.

Fig 2G: his figure is incomprehensible, mostly because the figure legend is cryptic. After some puzzling I figured it out, but many readers will not understand this. What is the dotted line? What is the vertical axis? How do the left and right panels exactly differ? Why are there no units on the O/E bars (and what is the point of these bars anyway)? Which cell type was analyzed?

Fig 3D: What is the explanation for the different correlation lengths for various PolII phosphorylation states? This potentially interesting finding should be discussed.

Fig 4C-F: The analyses are rather difficult to grasp, and it seems that overall the patterns are very weak. This part of the ms is rather tedious. Perhaps part or most of it can be left out?

MINOR:

Title: "underlies" suggests a causality that is not addressed here. Better would be "is linked to" or "correlates with".

Abstract 1st sentence mentions both "megabase" and "multi-megabase". Confusing.

p4, bottom paragraph is confusing: at first it seems to describe Hi-C data, but then it shifts to gene expression data.

The human Hi-C data were apparently analyzed for each chromosome arm separately. But it would be interesting to address whether the two arms of each chromosome represent the highest meta-TAD level. Or, in other words: do centromeres form strong boundaries that prevent metaTAD formation?

LAD overlap analysis is done with a metaTAD cutoff of 10Mb. Are the results different with different cutoffs?

p14, 2nd paragraph: please check the first sentence.

Supplementary Figures: They seem all very carefully constructed, but there are so many - this reviewer cannot possibly check all of them. Most readers may also welcome a smaller number. Please consider removing some of them, particularly if they merely repeat a main figure.

1st Revision - authors' response

11 November 2015

Thank you for considering our manuscript and for sending the Reviewer's comments. We believe we have addressed all of their concerns in our revised and much improved manuscript, as detailed in this letter. As you pointed out, the more fundamental issues of the Reviewers were the following:

- *Validate metaTADs by an independent method such as FISH (Reviewer #1).*

To answer this issue, we have performed new FISH experiments that measure the relationship between Hi-C interaction levels between genomically distant TADs and their physical proximity. Using FISH, we found that TADs that are distant along the DNA sequence but have strong Hi-C interactions (as in metaTADs) can indeed contact each other at short spatial distance

(TAD pairs more than 1Mb apart can achieve proximities of only 350 ± 242 nm). Thus, FISH confirms that distal TADs with high Hi-C interactions do enter into contact at short physical distances and form higher order topological structures (see new Figure EV5).

- *Add analyses of potential "binders" (Reviewer #2).*

We have expanded our analysis in mESCs to include a number of potential binders. Specifically, we examined the correlation lengths of twelve sequence-specific transcription factors mapped in mESCs by ChIP-seq (Chen et al. 2008 *Cell* 133: 1106). These include STAT3, Smad1, Zfx, c-Myc, n-Myc, Tcfcp2l1, E2f1 and the pluripotency factors Nanog, Oct4, Sox2, Klf4 and Esrrb. We found that all transcription factors in TAD pairs have longer correlation lengths in metaTAD trees than linear genomic distances, and strikingly this is also the case if one considers the pluripotency factors alone, confirming our previous observations with a range of other chromatin features. These new results are included in revised Fig 3.

- *Provide information on statistical support for our findings (Reviewer #1).*

Statistical support for all of our findings and information on our analyses were included in our original manuscript, although the information was sometimes only mentioned in figure legends and in the Appendix. To improve clarity, we have revised the manuscript throughout to clearly state where this information is detailed in the relevant Appendix sections, and highlighting the statistical methods and tests used in the Main Text.

- *Include additional controls and evidence of functional relevance (Reviewer #1).*

We have included new FISH analyses as detailed above. As requested (*Reviewer #1, specific point 1*), we have added a new supplementary figure illustrating the scale of interactions within and between metaTADs (Fig EV2). As suggested in *specific point 3* (and also recommended by Reviewer 2), we analyzed the relationship of metaTAD organization to additional chromatin features, some of which demonstrate intriguing correlations with metaTADs. Interestingly, we find that, e.g., the H3K9me3 histone mark characteristic of heterochromatin does not correlate across the metaTAD tree, more than expected (i.e. relative to linear genomic distance). In contrast many other features do show such correlations, such as those with roles in active transcription and active or facultative chromatin structures. Furthermore, as requested by Reviewer 1 (*specific point 8*), we have also improved our analyses in Fig 4, by including genes that do not change expression during differentiation.

We were reassured that Reviewers did not consider CRISPR/Cas9 experiments to be mandatory for the acceptance of our work. As Reviewer 2 observed, if we intended to disrupt a metaTAD structure using CRISPR/Cas9, it is currently unclear what sites within these multi-megabase structures should be targeted, and in any case, a comprehensive understanding of the functional relevance of metaTADs will require systematic approaches that will take years.

As requested, we have now modified the manuscript to include (1) a "standfirst text" summary, (2) three to four "bullet points" highlighting the main findings, and (3) a "thumbnail image" for the journal's homepage. We have also added a "Data availability" section. We have selected 5 figures to be included as "Expanded View" features and refer to them as Figure EVx in the Main Text. All other Supplementary Figures and Tables have been transferred to an Appendix and are referred to as Appendix Figure Sx or Table EVx. Finally, we have completed the author checklist, which we submit with our revised manuscript. Below are our detailed answers to the comments from both Reviewers.

Reviewer 1

We are pleased that the Reviewer deemed our notion of metaTADs "*in all likelihood correct*", and provide answers to her/his comments and suggestions below.

Specific points:

1. *The authors should provide a sense of scale for the frequency of TAD-TAD interactions at each level of the tree. The long-range interactions are not well visible in the maps. Please provide*

frequencies of interactions at each level. How frequent are TAD-TAD interactions relative to intra-TAD interactions, how frequent are second level metaTAD-metaTAD interactions etc.?

We thank the Reviewer for this suggestion and now include a new figure (Fig EV2), which gives a sense of scale for interactions within and between metaTAD domains. In this figure, we plotted the frequency of Hi-C interactions either within (*J*; purple) or between (*I*; blue) domains for ESC, NPC and Neurons. The figure shows that average interaction frequencies within domains progressively decrease as domains become larger in the three cell types. As expected, and in agreement with the steady decline in Hi-C contact frequency over genomic distance, the figure shows that interactions between domains also decrease with increasing domain size, but at a greater rate.

2a. *The authors correlate metaTADs with numerous chromatin features. It is difficult to know which ones of these, if any, are relevant and are the drivers, as the authors suggest.*

We agree that at this stage it is hard to pinpoint a set of features that drive the preferential contacts observed between TADs. Our observations that chromatin features related to active gene expression correlate with metaTAD structures, and that higher-order TAD contacts reorganize to maintain these correlations from pluripotency to terminal differentiation, support the view that metaTADs have functional roles. We have now expanded our analyses to include additional features, such as transcription factors and histone marks. In particular, we conclude that constitutive heterochromatin features do not explain the patterns of metaTAD organization seen. We agree that the correlations in Fig 3 do not identify specific drivers, and have revisited the text to make sure that such an impression is not given.

2b. *Some seem to contradict each other (presence of active RNA pol II and H3K27me3).*

Previous work has shown that chromatin marked by Polycomb mark H3K27me3 is also simultaneously bound by RNA pol II phosphorylated on Serine-5 residues in mouse ESC grown in serum/LIF (JK Stock et al. 2007 Nature Cell Biol.; E Brookes et al. 2012 Cell Stem Cell; Tee et al. 2014 Cell). It is therefore not contradictory, but consistent, that both RNAPII-S5p and H3K27me3 have similar correlation lengths across the metaTAD trees. Considering that RNAPII-S5p marks poised states of genes expression, both at active and Polycomb-repressed promoters, whereas S7p and S2p mark respectively active promoters and active coding regions, respectively, it is also interesting to see that the correlation is highest for S5p, lower for S7p and then smallest for S2p. Finally, such lower correlation length with S2p, nicely fits with a lower correlation length with CAGE data, which independently measures active transcription. We have added a short sentence to interpret these differences in the revised manuscript. We hope that our current investigation will inspire future studies seeking the drivers of chromatin contacts at different genomic scales, which will require new efforts.

3. *It is surprising that the authors only tested for a single histone modification (H3K27me3). Additional ones should be analyzed. Both correlating and non-correlating ones should be shown; particularly modifications related to transcriptional activity should be tested. Statistical analysis of their distribution relative to metaTADs should be provided.*

We thank the Reviewer for encouraging us to expand these analyses. We have now considered five additional histone marks related to variation in transcriptional activity including H3K4me3, H3K27ac, H3K36me3, H3K9me3, and H4K20me3. We found that modifications associated with active transcription (H3K4me3, H3K27ac, H3K36me3) have higher correlation lengths in metaTAD trees compared to linear genomic distances. In contrast, the H3K9me3 and H4K20me3 marks associated with constitutive heterochromatin do not display longer correlation lengths over the trees than those observed over linear genomic distances. That supports the view that long-range TAD contacts reflect similarities in chromatin features that also relate to transcriptional activity, or primed states of gene activation. We now refer to these results in the manuscript on page 11-12 and the new data are shown in our revised Fig 3, Fig EV3A, Appendix Fig S11-S12. The p-values depicting significantly increased correlations in comparisons to random neighbor trees can be found in Fig 3 and Appendix Fig S12.

4. *The conclusion that metaTADs are functionally important relies entirely on correlation.*

Can the authors disrupt metaTADs, for example, by altering some of the chromatin features they appear to correlate with (I realize the proper experiment of CRISPR disruption will probably be deemed "beyond the scope of this study"). In the absence of direct functional testing by experimentation, functional claims are not valid and should not be made.

We agree with the reviewer that it will be extremely interesting to understand the effects of genomic rearrangements (including deletions) to further understand long-range TAD contacts identified in our study. To this end, we are just starting to investigate how naturally occurring (disease-associated) TAD deletions affect metaTAD trees and gene expression. We agree with the reviewer that disruption studies are beyond the scope of the current manuscript, and will require substantial effort and significant time. As pointed out by Reviewer 2, it is also currently unclear what sites should be targeted for disruption and what consequences would be expected (i.e. what features of genome biology should be measured to test the effectiveness of metaTAD disruption). We feel that questions on the functional relevance of metaTAD organization will be best addressed using careful systematic approaches that will likely take years.

We have carefully revised the text with the aim of removing unjustified claims for the functional importance of metaTADs. However we think that it is fair to conclude from our analyses, that metaTAD contacts reflect specific functional states (e.g. states of gene expression), and are not a simple reflection of DNA sequence features or genomic length. We have sought to strengthen this perspective in the revised version of the manuscript, and are grateful for the Reviewers' comments prompting our study to be clearer and more robust. In a similar spirit, we have revised the Title substituting "underlie" to "are linked to".

5. *The authors should validate metaTADs by an independent method such as FISH and show differential pairing in ESC, NPC and neurons as proposed in figure 6, including of genes which change activity during differentiation and switch metaTADs as suggested by the authors in figure 6.*

As FISH experiments are very time-consuming, to answer this request in a timely fashion we have examined the behavior of a set of probes at the *Bmp7* locus, which were readily available in our laboratories. We considered three probes, *a* (red), *b* (green) and *c* (blue), belonging to non-contiguous TADs, separated by similar genomic distances (1.5Mb for *a-b* and 2Mb for *b-c*, respectively). The probe positions are such that they correspond to regions in the Hi-C matrix that have noticeably different degrees of interaction; interaction frequency between *a* and *b* is ~90% higher than between *b* and *c*. We found that the distant pair of TADs *a* and *b* contact each other at short spatial distance, as their centers of mass are separated by only 350 ± 242 nm. This distance is significantly closer than the distance that separates loci *b* and *c* (587 ± 375 nm; Wilcoxon test p-value 0.00168). The ratio of their physical distances ($b-c/a-b$) is 30% larger than their genomic distance ratio. Thus, FISH confirms that distal TADs sharing strong Hi-C interactions (as seen in metaTADs) contact at shorter physical distances than TAD pairs separated by a similar linear genomic distance but sharing weaker Hi-C interactions.

We also checked whether validation could be conducted with published FISH data. We examined, for instance, the 2D FISH data from Dixon et al. Nature 2012 (two distinct regions), and 3D FISH datasets from Williamson et al. Genes & Dev 2014 (eight different datasets). Unfortunately, all probes in these datasets were contained within a single TAD and could not be used to show interactions across TADs that form metaTADs.

6. *The authors find conservation of TADs between 74-80% in pairwise comparisons between ESC, NPC and neurons. How does this compare to an entirely unrelated cell line, say fibroblasts? Is this level of conservation functionally important?*

We have now examined Hi-C data from mouse fibroblasts published recently by Battulin N. et al. 2015 Genome Biology. Unfortunately, substantially lower TAD numbers were identified in that study compared to our datasets (~3,250 vs 5,250), most likely due to the shallow sequencing depth (around half of the sequencing depth we achieved) for the fibroblast data. Nevertheless, we find that ~75-82% of the TAD boundaries identified in fibroblasts are conserved in our three datasets. We also examined the conservation of fibroblast TAD boundaries against the original ESC-J1 Hi-C dataset published by the Ren group (Dixon et al. 2012), which features a similar number of boundaries (ESC-J1 (HMM) gives 3,127 TADs). The conservation between these two datasets is 66%.

We have not added the comparison with fibroblasts to our revised manuscript, due to the lower sequencing depth of the fibroblast datasets, but we are willing to do so if the reviewer feels it is essential. The additional fibroblast column could be added to the current Appendix Fig 6. Whether conservation is functionally important is still debated, but recent work investigating specific boundaries conserved between human and mouse suggests so (see, e.g., Lupianez et al. Cell 2015).

7. *A significant overlap between regions with both tree change and gene expression change is found in the ESC-Neurons transition, both in compartment A or regions that undergo A/B compartment changes". How is significance determined in this analysis? What are the controls? What are the statistical methods used and what are the values of statistical comparators? The authors use the argument that genes which change expression during differentiation are found in changing metaTADs. Please provide comprehensive controls of genes which do not change expression during differentiation.*

The previous version of the manuscript described statistical methods, significance of tests (e.g., p-values) and controls for all of our findings within the SI Appendix. We sympathize with the Reviewer's request for clarity, and to this end we have modified the Main Text accordingly to highlight the significance of our statistical tests and our methods.

We thank the Reviewer for the suggestion to test our finding that "*genes which change expression during differentiation are found in changing metaTADs*" against a "*control of TADs which do not change expression during differentiation*". The new analyses are included in Fig 4 and in Appendix Fig 17, and the results reinforce the specificity and strength of our previous results. In our revision, we also realized that Fig 4 was hard to grasp, and we have made an effort to illustrate our analyses more clearly and to simplify both Main Text and Fig 4, as we summarize below.

To link local gene expression changes with local tree topology changes and A/B compartment membership changes, we first identified genomic intervals with TADs exhibiting significant changes. We used standard statistical methods that are briefly detailed below, fully explained in Appendix Supplementary Methods (sub-heading *Local tree changes, expression changes and A/B compartment membership changes*) and stated in Appendix Figs S16-S17. We then calculated the overlaps between genomic intervals having those different features, using the Jaccard Index which is a standard measure of overlap (Fig 4E and Appendix Supplementary Methods) that is now graphically summarized in our revised Fig 4F.

As detailed in Appendix Supplementary Methods, the significance of overlaps was assessed by comparing the overlap value obtained from the real biological data with the overlaps obtained from a set of equivalent genomic intervals that were randomly placed in the genome. We performed 10,000 circular random permutations, whereby the random datasets have intervals with the same length and genomic clustering patterns as the real data, but the locations of the intervals is randomly permuted. If the overlap between two sets of intervals is associated with a low permutation derived p-value, we conclude that the features compared coincide in the same genomic regions more than can be expected by chance.

Prompted by the Reviewer's request for the use of a control set, we now include comparisons with regions of the genome that do not undergo changes in gene expression, and examine the associated tree and compartment changes (in revised Fig 4F). We show that regions with no gene expression changes are invariably found within genomic intervals that always remain in compartment B ('repressed-chromatin'). Interestingly, the association of 'no gene expression change' with B compartment is true both in areas of the tree with conserved topology, but also in regions of the trees that change topology.

Taken together, our results show that changes in gene expression during differentiation are mostly found in compartment A, or in A-B/B-A compartment changes. This pattern, interestingly, occurs both in areas of the tree that change local topology and in areas of the tree that maintain topology. The unexpected finding that changes of gene expression can be associated with regions with conserved local tree topology is explained by the observation that the changes in gene expression within conserved metaTADs tend to be coherent (Fig 4G): in other words, when both TADs in a contacting pair coherently alter or maintain expression, they remain together in a

conserved metaTAD. Finally, regions of the genome that do not change expression tend to be and to remain in compartment B, yet interestingly such regions can also rewire their TAD contacts. A speculation is that such contact restructuring may be linked to topological rewiring associated with gene expression changes occurring elsewhere.

8. *There is a general lack of information on the statistical analysis used. Numerous statements regarding significance and differences are made without controls or indication of statistical power. Please go through the manuscript and provide indicators of statistical significance of all claimed differences.*

As explained before, statistical methods, controls, and information on statistical analyses were featured in the first version of our manuscript, particularly in the Supplementary Information file and figure legends, but also featured in the Main Text. We realize that the manuscript was not clear enough about where these items could be found. We have now gone through the manuscript to more clearly point out where methods, information, controls and indicators of statistical significance are described. We have also tried to specify the most relevant information here in the resubmission letter.

Minor points:

1. *The authors state that interaction strength is measured (p. 6). This is not the case. What is measured is an interaction signal. What this signal means is not known or tested. Please rephrase.*

We have changed the word "strongly" to "frequently" in the first paragraph, and replaced "stronger" with "more frequent" in the second. The remaining text was also adjusted.

2. *With regards to nomenclature, would it be useful to introduce the notion of metaTAD-I, metaTAD-II etc. to indicate where in the tree a metaTAD is.*

We thank the Reviewer and introduced this notion on page 6 of the Main Text.

3. *The participating authors from the FANTOM consortium should be listed somewhere.*

We apologize for the confusion, but we were following the FANTOM5 guidelines for authorship specification. All contributing authors from the FANTOM consortium were already listed individually in the original authorship list. The FANTOM5 consortium requires that all papers using data produced in the context of the FANTOM project (here the CAGE data produced at RIKEN from samples generated in the Pombo lab) should include an additional author indicated as "the FANTOM Consortium".

Reviewer 2

We thank the Reviewer for her/his helpful and positive report, stating that "a hierarchical organization [of chromatin] has been noted before, but [...] not been captured in a computational framework yet", that our study is "interesting and timely" and "will fit well in MSB". Below are our answers to her/his comments and suggestions.

Major points:

1. *SBS modeling (Figure 5): interesting, but it remains very theoretical. A key question is: what are the binders? Here I wonder whether an opportunity has been missed. Figure 3 only compares metaTAD architecture to chromatin marks that are unlikely to be binders. But what about DNA-binding factors and their sequence motifs (e.g. Jaspar database): do any of these show a TAD/metaTAD enrichment pattern? For a few human cell types there is a lot of ENCODE data as well as Hi-C data to test this.*

We thank the Reviewer for his/her interest in the SBS modeling component of the manuscript and the suggestion to expand our analysis to include transcription factors. We have expanded our analyses relating with Fig 3 to examine the correlations over the metaTAD tree of 12 sequence-specific transcription factors mapped in mESCs by ChIP-seq (Chen et al. 2008 *Cell* 133: 1106). These include STAT3, Smad1, Zfx, c-Myc, n-Myc, Tefcp211, E2f1, and the pluripotency factors Nanog, Oct4, Sox2, Klf4, and Esrrb. We found that like CTCF, several transcription factors analyzed have longer correlation lengths in metaTAD trees as compared to linear genomic distances. Indeed, correlation lengths extended over twice as far along the trees than along the linear sequence when either the pluripotency factors were considered alone, or when all TFs were included. We refer to these new results on pages 11-13 of the manuscript, in Fig 3D and, with more details, in Appendix Fig S11-12. The p-values for the statistical significance of the correlations found are given in Appendix Fig S12.

Following a similar suggestion from Reviewer 1 to look at additional histone marks, we found that histone marks associated with active (e.g. H3K4me3 and H3K36me3) and poised (H3K4me3 and H3K27me3) states of gene expression also have higher correlation lengths across trees than linear genomic distances, but remarkably that histone marks associated with constitutive heterochromatin (H3K9me3 and H3K20me3) do not. Considered together, our results point to a scenario where states of gene activity (which are cell-type specific) contribute to long-range contacts across genomic length scales, but due to the complexity of binding combinations (plus the large number of binding factors with no available ChIP-seq data) it is difficult to pinpoint a single factor (or small number of factors) that determine folding. It is also possible that different factors have different contributions at different genomic scales. It is still too early to provide detailed answers to questions about the nature of candidate binders, yet we are using the SBS model in separate studies to explore exactly this topic, and we hope that our current work will encourage further research.

2. *In addition: Surely the SBS model is not the only model that may explain metaTADs. Needs to be addressed in the Discussion.*

We thank the Reviewer for this suggestion and have now expanded our discussion on this topic on pages 19-20 as allowed by manuscript length constraints. The SBS model is the first to take into account metaTADs and has the advantage of explaining within a single framework a variety of data, from Hi-C and FISH (see Barbieri M. et al. 2012 PNAS). As a comparison, a simple Self-Avoiding Walk polymer model or the Fractal Globule model would predict homogeneous Hi-C contact matrices with no TADs or metaTADs. Conversely, other models such as those considering the effects of supercoiling and plectonemes could also be relevant (see our review in *Curr. Opin. Cell Bio.* 2014).

3. *Fig 2G: this figure is incomprehensible, mostly because the figure legend is cryptic. After some puzzling I figured it out, but many readers will not understand this. What is the dotted line? What is the vertical axis? How do the left and right panels exactly differ? Why are there no units on the O/E bars (and what is the point of these bars anyway)? Which cell type was analyzed?*

Thank you for pointing out issues with panel 2G. We apologize for the brief, unclear legend and now provide a better description of Fig 2G, and a simplified panel, which we hope improves clarity.

4. *Fig 3D: What is the explanation for the different correlation lengths for various Pol II phosphorylation states? This potentially interesting finding should be discussed.*

Previous work from the Pombo and Reinberg labs has shown that, in mouse embryonic stem cells, RNA pol II phosphorylated on Serine-5 residues marks both the promoters and coding regions of active genes, as well as the ~4000 cohort of Polycomb-repressed genes which are marked by H3K27me3 (JK Stock et al. 2007 *Nature Cell Biol.*; E Brookes et al. 2012 *Cell Stem Cell*; Tee et al. 2014 *Cell*). In contrast, RNA pol II phosphorylated on S7p abundantly marks the promoters of active genes, whereas S2p is found at apparently lower levels throughout the coding regions of productively expressed genes. It is therefore quite striking, as the reviewer points out, that we see S5p having a larger correlation length (which perhaps not accidentally is similar to H3K4me3, and to H3K27me3), followed by S7p and S2p, with matching lower correlation lengths from the histone

marks related with productive elongation (H3K36me3). Furthermore, the lower correlation length with S2p nicely fits with a lower correlation length with CAGE data, which measures active transcription. As mentioned above, we have added a short sentence on this topic to the manuscript to aid the interpretation of the results, although we feel strongly that the current analyses aim to showcase opportunities for more detailed analyses to be explored in the future with additional datasets.

5. *Fig 4C-F: The analyses are rather difficult to grasp, and it seems that overall the patterns are very weak. This part of the ms is rather tedious. Perhaps part or most of it can be left out?*

As also suggest by Reviewer 1, we have re-worked and expanded significantly the analyses in Fig 4, re-shaped the content of that figure to make it clearer and more accessible, and improved the text. The analyses presented in this figure are very complex because they compare changes during differentiation in three variables (tree of contacts, gene expression and compartments A/B). Nevertheless, we feel that these analyses provide interesting results, which are worthwhile presenting as they help to test and explore the possible functional relevance of the metaTAD structure and its impact on gene expression.

As written in our reply to Reviewer 1, taken together, our results show that changes in gene expression during differentiation can be found in compartment A, or in A-B/B-A compartment changes. That pattern, interestingly, occurs both in areas of the tree that change local topology and in areas of the tree that maintain topology. We were not expecting to find an association between changes of gene expression with regions with conserved local tree topology. However this observation raised one remarkable possibility, that if both distant TADs changed expression in a coherent manner, they would still remain associated. This was indeed the result (shown Fig 4G). In the new analyses suggested by Reviewer 1, where we further test what happens to TADs without gene expression changes, we find that they tend to be and remain in compartment B, as expected, but interestingly these regions can also rewire their TAD contacts. We speculate that the restructuring of such contacts could be indirectly linked to topological rewiring associated with gene expression changes occurring elsewhere. This complex question will be the focus of extensive future work.

Our analyses so far are all consistent with the existence of long-range TAD contacts that relate to transcriptionally activated chromatin states, and to the rewiring of contacts to preserve like-with-like contacts upon a change in gene expression (as occurs during differentiation). Many more studies will be necessary to fully explore the details and functional implications of such contacts, but we feel that our manuscript opens the way to this exploration.

Minor points:

1. *Title: "underlies" suggests a causality that is not addressed here. Better would be "is linked to" or "correlates with".*

We modified the title in response to the Reviewer's comment as follows: "Hierarchical folding and reorganization of chromosomes are linked to transcriptional changes in cellular differentiation"

2. *Abstract 1st sentence mentions both "megabase" and "multi-megabase". Confusing.*

We agree with the Reviewer and modified the first sentence of the Abstract as follows: "Mammalian chromosomes fold into arrays of megabase-sized topologically associating domains (TADs), which are arranged into compartments spanning tens of megabases of genomic DNA."

3. *p4, bottom paragraph is confusing: at first it seems to describe Hi-C data, but then it shifts to gene expression data.*

We have modified the indicated paragraph on page 4 and 5 to clarify this point.

4. *The human Hi-C data were apparently analyzed for each chromosome arm separately. But it would be interesting to address whether the two arms of each chromosome represent the highest*

meta-TAD level. Or, in other words: do centromeres form strong boundaries that prevent metaTAD formation?

We thank the Reviewer for this very interesting question. Most of our work was done with mouse (acrocentric) chromosomes. To follow the reviewer's suggestion, we repeated our clustering analysis using the human ESC-H1 and IMR90 Hi-C datasets, allowing the metaTAD discovery to extend across the whole chromosomes, to capture interactions across chromosomal arms. We found that tree structures in each arm remained totally unaltered for all chromosomes in both cell lines, with the exception of chr1 and chr9 in ESC-H1 where nevertheless the conservation remained >97%, and the differences are minor and observed only at the top levels of the tree. Hence, if we consider whole chromosomes, the centromere is the boundary at the very highest level of the tree, or as suggested by the Reviewer, centromeres do appear to represent boundaries between the arms that prevent their interactions. However, we find that metaTAD signals at scales comparable with the size of entire chromosomal arms become comparable to the background signal (see Fig 2D) hence metaTADs cannot be identified with acceptable statistical confidence.

5. *LAD overlap analysis is done with a metaTAD cutoff of 10Mb. Are the results different with different cutoffs?*

The results do not significantly differ when different metaTAD cutoffs are selected. We have added a new supplementary figure (Appendix Fig S10) which shows similar results are obtained when selecting cutoffs of 5, 10 or 20Mb.

6. *p14, 2nd paragraph: please check the first sentence.*

We thank the Reviewer and have corrected that sentence.

7. *Supplementary Figures: They seem all very carefully constructed, but there are so many - this reviewer cannot possibly check all of them. Most readers may also welcome a smaller number. Please consider removing some of them, particularly if they merely repeat a main figure.*

We thank the Reviewer for the feedback. We opted to keep our Supplementary Figures for the sake of completeness, but we have simplified the organization and presentation of the materials in a manner adhering to *Molecular Systems Biology's* guidelines. MSB has replaced "Supplementary Information" with an Expanded View figure section that can be directly accessed by readers online in the Main Text. The rest of the Supplementary Figures as well as the Supplementary Methods and Analyses, which may benefit those involved in similar types of studies, were included in a more traditional Appendix".

2nd Editorial Decision

24 November 2015

Thank you again for submitting your work to Molecular Systems Biology. We have now heard back from the two referees who agreed to evaluate your revised manuscript. As you will see below, the referees are satisfied with the modifications made and they think that the study is now suitable for publication.

Before we formally accept the manuscript, we would like to ask you to address the editorial issues listed below.

Reviewer #1:

The authors have done an exemplary job in this revision. In particular, they have removed over-reaching statements. This manuscript reports an interesting novel concept and I support its publication.

Reviewer #2:

The authors have improved the manuscript substantially. I now strongly recommend publication in MSB.